# Profiling of the Bacterial Microbiota along the Murine Alimentary Tract

**DOI:** 10.3390/ijms23031783

**Published:** 2022-02-04

**Authors:** Ramiro Vilchez-Vargas, Franz Salm, Eva B. Znalesniak, Katharina Haupenthal, Denny Schanze, Martin Zenker, Alexander Link, Werner Hoffmann

**Affiliations:** 1Department of Gastroenterology, Hepatology, and Infectiology, Otto-von-Guericke University Magdeburg, Leipziger Str. 44, 39120 Magdeburg, Germany; ramiro.vilchez@med.ovgu.de; 2Institute of Molecular Biology and Medicinal Chemistry, Otto-von-Guericke University Magdeburg, Leipziger Str. 44, 39120 Magdeburg, Germany; franz.salm@gmx.de (F.S.); eva.znalesniak@med.ovgu.de (E.B.Z.); katharina.haupenthal@med.ovgu.de (K.H.); 3Institute of Human Genetics, Otto-von-Guericke University Magdeburg, Leipziger Str. 44, 39120 Magdeburg, Germany; denny.schanze@med.ovgu.de (D.S.); martin.zenker@med.ovgu.de (M.Z.)

**Keywords:** microbiota, gut bacteria, gut microbiome, trefoil factor, TFF3, mucus, stomach, intestine, esophagus, colitis

## Abstract

Here, the spatial distribution of the bacterial flora along the murine alimentary tract was evaluated using high throughput sequencing in wild-type and *Tff3*-deficient (*Tff3*^KO^) animals. Loss of *Tff3* was linked to increased dextran sodium sulfate-induced colitis. This systematic study shows the results of 13 different regions from the esophagus to the rectum. The number of bacterial species (richness) increased from the esophagus to the rectum, from 50 to 200, respectively. Additionally, the bacterial community structure changed continuously; the highest changes were between the upper/middle and lower gastrointestinal compartments when comparing adjacent regions. *Lactobacillus* was the major colonizer in the upper/middle gastrointestinal tract, especially in the esophagus and stomach. From the caecum, a drastic diminution of *Lactobacillus* occurred, while members of Lachnospiraceae significantly increased. A significant change occurred in the bacterial community between the ascending and the transverse colon with Bacteroidetes being the major colonizers with relative constant abundance until the rectum. Interestingly, wild-type and *Tff3*^KO^ animals did not show significant differences in their bacterial communities, suggesting that Tff3 is not involved in alterations of intraluminal or adhesive microbiota but is obviously important for mucosal protection, e.g., of the sensitive stem cells in the colonic crypts probably by a mucus plume.

## 1. Introduction

Mucous epithelia mainly cover the inner surfaces of the body and are essential for vital interactions with our environment (respiration, nutrition, reproduction, visual and auditory systems). As a consequence, the various mucosae represent a preferred location for microbiota. The microbial composition varies dramatically at different body sites in healthy humans [1]. Within the last decade, it has become increasingly clear that the microbiota has severe and diverse influences on health and disease, particularly in the gut [2]. On the one hand, diet is an important factor shaping microbial composition [3,4,5,6]. On the other hand, the gut microbiome impacts obesity and metabolic dysfunctions, such as metabolic syndrome, type 2 diabetes, and inflammation, as well as neurological disorders [7,8,9]. For example, bacterial metabolites (postbiotics) regulate gene expression, e.g., of mucins [6]. Furthermore, there are molecular links between microbiota and genotoxic events leading to various types of human cancers [10,11]. Thus, it is of increasing clinical interest that fecal microbiota transplants show therapeutic effects and there are many possible applications, such as inflammatory bowel disease, metabolic syndrome, obesity, autism, multiple sclerosis, Parkinson’s disease, and even cancer [6,12].

Only the development of culture-independent methods allowed a more complete identification of the complex microbiota. In the stomach, five major phyla (divisions) of bacteria were detected, i.e., Firmicutes, Bacteroidetes, Actinobacteria, Fusobacteria and Proteobacteria [13,14]; whereas the human distal gut microbiota is typically dominated by two bacterial phyla, i.e., the Firmicutes and the Bacteroidetes, and one member of the Archaea [15,16]. Generally, the majority of the microbes reside in the colon.

An essential element of the host–microbiota symbiosis is the mucus barrier, which typically covers mucous epithelia [13,17,18,19]. Characteristic components of these barriers are gel-forming mucins and their composition varies for each mucosa fulfilling a special physiological function [17]. Further components are various ions, and a complex mixture of additional proteins, such as immunoglobulins, trefoil factor family (TFF) peptides, gastrokines, IgG-Fc-binding protein (FCGBP), DMBT1/gp340, galectins, defensins, and lysozyme [17,20]. In addition, also the physical structure of the mucus layer varies along the gastrointestinal tract. A two-layered structure is found in the stomach and colon; whereas in the small intestine, the mucus layer is simpler [21]. On the one hand, the mucus layer is the first-line defense of the epithelium, which interacts with the immune system [22]. On the other hand, a dense microbial community is embedded in the mucus layer resembling a biofilm [19]. For example, colonic mucus encapsulates the fecal material including the microbiota, which direct their own encapsulation [23]. Upon induction of mucus defects, the bacterial loads increased, particularly in areas between fecal pellets, leading to inflammation [23,24].

In the past, mice were intensely used as a model system for studying various physiological effects of microbiota [19,25,26,27]. Our group has recently provided systematic data on microbial distribution through the human gastrointestinal (GI) tract [28]. However, the data are limited by an endoscopic biopsy collection, and potential bacterial cross-contamination cannot be excluded, in particular esophagus versus stomach versus duodenum or ileum versus colon versus rectum. Thus, we aimed to systematically analyze the bacterial microbiota along the alimentary tract of mice, which would allow to exclude potential endoscopy-related contamination.

In addition, the microbiota was also analyzed in *Tff3*-deficient (*Tff3*^KO^) mice. These mice were chosen as they react much more sensitively in the dextran sulfate sodium (DSS)-induced colitis model, probably due to an intestinal barrier defect [29,30]. In the DSS model, the thickness of the inner mucus layer decreases and becomes permeable to bacteria [31]. TFF3 is a member of the trefoil factor family [32,33] and forms a disulfide-linked heterodimer with FCGBP [34,35]. TFF3, together with FCGBP and the mucin MUC2, is predominantly secreted from intestinal goblet cells (GCs) [34,36,37]. Minor amounts of TFF3 and FCGBP are synthesized along the alimentary tract, i.e., in labial and submandibular glands [38,39], in esophageal submucosal glands and the gastric cardia [40], as well as in the gastric antrum [41,42,43]. FCGBP and TFF3-FCGBP are considered components of the mucosal innate immune defense regulating pathogen attachment and clearing of microorganisms [32,44,45]. This may be in connection with DMBT1/gp340, as the latter also binds to TFF3 [46]. DMBT1/gp340 is highest expressed in the intestine (by enterocytes [47]) and is identical to salivary agglutinin, which aggregates *Streptococcus* and agglutinates influenza A virus [48,49].

In the present study, we systematically characterized the bacterial communities from 13 different locations along the alimentary/GI tract from wild-type and *Tff3*^KO^ mice with the aim of testing whether a lack of Tff3 affects the composition of the microbiota, particularly in the intestine. This investigation was also designed to establish a detailed topographical distribution of the bacterial microbiota throughout the entire murine GI tract, which is a prerequisite to studying bacterial communities in different mouse models, such as probiotic trials. Of note, probiotics upregulate Tff1 and Tff2 in the murine stomach [50].

## 2. Results

Bacterial communities were analyzed throughout the alimentary tract of nine wild-type and nine *Tff3*^KO^ mice. The following anatomical locations were investigated: esophagus; forestomach, gastric corpus, gastric antrum; proximal and distal parts of the duodenum, respectively; the middle section of the jejunum; distal ileum; the middle part of the caecum; proximal (ascending), medial (transverse), and distal (descending) colon; rectum. Generally, the microbiota in the luminal content, as well as the mucus, were analyzed altogether without differentiation. After resampling to the minimum sequencing depth of 3172 reads per sample, a total of 624,884 sequences were retrieved from 197 samples successfully sequenced (37 samples, mainly from the lower GI tract, did not yield any PCR product). A total of 4560 phylotypes were detected in the whole cohort, 4297 phylotypes were assigned to a phylum (263 remained as unclassified bacteria), 3492 were assigned to family and 2453 were assigned to the genus (Appendix A: raw data with the phyloptypes sequences).

### 2.1. Overall Bacterial Communities in Wild-Type and Tff3^KO^ Mice

In both groups of mice, there was an increase in the species richness from the esophagus to the rectum. For example, in the esophagus the species richness was about 50; in the stomach, duodenum, jejunum and ileum it was roughly 100; whereas in the caecum, colon and rectum it increased to about 200. In line with this, the diversity also increased throughout the gastrointestinal tract and reached a maximum in the distal part of the intestine, as measured by the Simpson index (Figure 1).

Both groups of mice (wild-type and *Tff3*^KO^) did not show statistically significant differences with respect to the richness and diversity of the bacterial communities when the corresponding anatomical regions were compared pairwise (*p*-value > 0.05; data not illustrated). This suggests that the lack of Tff3 did not affect the relative abundance of bacterial taxa throughout the entire gastrointestinal tract. 

### 2.2. Spatial Analysis of the Bacterial Communities at the Genus Level

The composition of the bacterial communities showed a clear shift between the ileum and the caecum in both the wild-type (Figure 2; *p* < 0.001) and *Tff3*^KO^ mice (Figure 3; *p* < 0.05 in *Tff3*^KO^ mice). The genus *Lactobacillus* was the major colonizer from the esophagus to the ileum, including the stomach, duodenum and jejunum. In the esophagus, *Lactobacillus* made up even more than 95% of the total bacterial communities. Particularly in the caecum and ascending colon, the relative abundance of the phylum Firmicutes dropped due to a drastic decrease in *Lactobacillus* (*p* < 0.01), the latter was replaced by other taxa belonging to Firmicutes (Lachnospiraceae, unclassified Clostridiales, and unclassified Ruminococcaceae) as well as Deferribacteres (*Mucispirillum*). Furthermore, members of the phylum Bacteroidetes (*Alistipes*) appeared and were the natural colonizers from the caecum to the rectum. Interestingly, PERMANOVA also revealed differences in the overall bacterial communities between the proximal (ascending) part of the colon and the transverse colon in both the wild-type (Figure 2) and *Tff3*^KO^ mice (Figure 3). 

Thus, three regions could be distinguished along the murine alimentary tract based on their bacterial communities. First, the esophagus, stomach, duodenum, jejunum, and ileum are dominated by *Lactobacillus* as the major colonizer with a progressive enrichment of the phylum Bacteroidetes (unclassified Bacteroidales and unclassified Bacteroidetes). The second region includes the caecum and the proximal (ascending) colon; whereas, the medial (transverse) colon, the distal (descending) colon and the rectum represent the third characteristic region.

As a consequence of the systematic sampling methodology, no statistically significant differences were found between adjacent regions, with the exception of ileum versus caecum and ascending versus transverse colon, as mentioned above. However, a systematic pairwise comparison of all the regions analyzed revealed that the bacterial communities progressively changed from the esophagus to the rectum (Appendix A: pairwise comparisons). For instance, no differences between the esophagus and forestomach and corpus were detected, but bacterial communities between the esophagus and antrum were significantly different (*p* < 0.05); these differences became even more significant when comparing the esophagus with the duodenum or the other more distal regions (*p* < 0.001). Interestingly, pairwise comparisons between bacterial communities of the proximal duodenum and distal duodenum or jejunum or ileum did not show statistically significant differences. This indicates that the microbiota is stable from the proximal duodenum to the ileum; whereas, there is a statistically significant difference starting with the caecum (Appendix A: pairwise comparisons).

Pairwise comparison of the corresponding anatomical regions of both groups of mice (wild-type and *Tff3*^KO^) did not reveal statistically significant differences in most regions with respect to the composition of the bacterial communities (*p*-value > 0.05; data not illustrated). The only exception was the ascending colon, which significantly differed at the genus level (*p* = 0.02; data not illustrated). This suggests that a lack of Tff3 did not affect the relative abundance of bacterial taxa in most regions of the alimentary tract.

### 2.3. Spatial Analysis of Lactobacillus Phylotypes

When analyzing the sequences at the maximum level of resolution, i.e., the phylotype level, it became apparent that the bacterial communities of the esophagus, in both the wild-type and *Tff3*^KO^ mice clustered separately (Figure 4). A more detailed analysis revealed that the esophagus was colonized by phylotypes belonging to *Lactobacillus murinus* (Phy2, Phy21, Phy29 among others); whereas phylotypes belonging to *Lactobacillus reuteri* (Phy4, Phy7 and Phy19) and *Lactobacillus gasseri/taiwanensis* (Phy1, Phy11 and Phy59) were detected from the stomach to the ileum, *L. murinus* being almost absent in these regions (Figure 5). Clearly, three different regions could be distinguished in both the wild-type and *Tff3*^KO^ mice: first, the esophagus grouped separately and is characterized by the presence of *L. murinus*; second, a region spanning from the forestomach until the ileum, where *L. gasseri/taiwanensis*, *L. reuteri*, and only a little *L. murinus* were detectable; third, the lower GI tract from the caecum to the rectum, where *L. murinus* is missing and *L. gasseri/taiwanensis* and *L. reuteri* were hardly detectable. Here, after the ileum, the bacterial community showed the main profile change (Figure 5). Furthermore, there were no significant differences detectable between wild-type and *Tff3*^KO^ animals.

## 3. Discussion

Here, we provide a comprehensive analysis of the bacterial communities in the murine alimentary tract by investigating 13 different anatomical regions from the esophagus to the rectum. Furthermore, *Tff3*^KO^ mice were compared with wild-type littermates as *Tff3*^KO^ mice react much more sensitively in the DSS colitis model, probably due to a defect in the colonic inner mucus barrier [29,30]. Here, Tff3 together with Tff3-Fcgbp is part of the mucosal innate immune defense, presumably regulating microbial attachment [32,44,45]. Of note, Tff3 expression is also linked to inflammation (for review, see [30]) and goblet cells are subject to complex immunomodulation. For example, Tff3 expression is reduced by the immune system after infection with *Citrobacter rodentium*, which is not observed in *Rag1*^KO^ mice [51]. This could be explained by a link of T lymphocytes and the homeostasis of intestinal epithelial cells by IL-7 [52]. Furthermore, Tff3 expression is induced by Toll-like receptor 2 (TLR2) activation by commensal bacteria, probably by an indirect mechanism [53].

### 3.1. The Bacterial Communities Differ Significantly along the Murine Alimentary Tract

Grossly, at the genus level, at least three regions were clearly distinguishable by their bacterial composition in both mouse strains investigated, with statistically significant differences between the distal ileum and the caecum, and also between the ascending colon and the transverse colon (Figure 2 and Figure 3). This indicates that there are at least three different ecosystems, which clearly reflect distinct biochemical and physiological conditions.

#### 3.1.1. The Upper Alimentary Tract

The upper alimentary tract from the esophagus to the distal ileum is dominated by *Lactobacillus* with decreasing abundance; whereas unclassified Bacteroidetes, unclassified Bacteroidales, and unclassified Clostridiales gradually increase (Figure 2 and Figure 3). Generally, the richness and the Simpson index continuously increased in the upper alimentary tract towards the ileum (Figure 1). For example, *Lactobacillus* contributes to about 95% of the bacterial abundance in the esophagus (Figure 2 and Figure 3), which probably originates from the saliva. Remarkably, on the phylotype level, the esophagus represents a distinct ecological niche as only here the aerotolerant anaerobe *L. murinus* was detected (Figure 5). In contrast, *L. gasseri/taiwanensis* and also *L. reuteri* were present in the forestomach to the ileum but were missing in the esophagus (Figure 4). Generally, the preferred habitat for *L. gasseri/taiwanensis* and *L. reuteri* is the stomach (Figure 4). Particularly for *L. reuteri* it was well documented that the adaptation to the stomach is also accompanied by a host specialization; for example, rodent-specific genes were inactivated in *L. reuteri* strains from different vertebrate hosts [54]. As a consequence, the ability to form epithelial biofilms in the murine forestomach is strictly dependent on the strain’s host origin [55]. Probably a similar specialization of *L. murinus* occurred to the esophageal epithelial cells (Figure 4). Of note, in a gnotobiotic mouse model colonized with eight strains of the altered Schaedler flora, the spatial distribution of *L. murinus* was very different from that presented here [56]. *L. murinus* was again present in the esophagus and absent in the non-glandular forestomach, but it declined again in the stomach until the ascending colon [56]. This clearly outlines the extreme sensibility of the microbial ecosystem and the enormous influence of the other members of the bacterial community.

#### 3.1.2. The Lower Alimentary Tract

Generally, the terminal ileum seems to be a major demarcation concerning microaerobic (*Lactobacillus*) in the small intestine and anaerobic genera in the large intestine, starting with the caecum (lower alimentary tract). There is also a step up in the richness (amount of different phylotypes) between the terminal ileum and the caecum (Figure 1). Starting with the caecum, obligate anaerobic bacteria are present nearly exclusively indicating that here there is no selection pressure anymore to tolerate oxygen in the atmosphere. Thus, the severe loss of oxygen distally of the terminal ileum (anaerobic environment in the caecum and colon) seems to be a major checkpoint for the selection of the bacterial microbiota. Furthermore, in both the wild-type and *Tff3*^KO^ mice, there is a highly significant difference between the bacterial communities in the ascending colon and the transverse colon (Figure 2 and Figure 3). For example, unclassified Bacteroidales and unclassified Bacteroidetes increase distally to the ascending colon, whereas unclassified Clostridiales decrease and *Mucispirillum* is strongly reduced. Thus, the caecum and the ascending colon represent a location with a special bacterial community when compared with the adjacent proximal and distal regions.

The characteristic bacterial communities in the caecum and the ascending colon are probably the result of an adaptation to the special physiological function and biochemical reactions typical of the lower alimentary tract, as well as certain anatomical features. Typically, in the proximal part of the large intestine, microbial degradation of special polysaccharides occurs (e.g., resistant starch), and it was compared with a bioreactor [57]. In humans, dietary plant fibers, which are composed of complex cell wall polysaccharides, are resistant to digestion and resorption in the small intestine [58]. For their breakdown, microbial fermentation in the large intestine is essential and this process provides about 10% of the human diet’s energy. Major end products of this fermentation are short-chain fatty acids, i.e., acetate, propionate, and butyrate; the latter being a major energy source for colonocytes [58]. 

In contrast, the distal colon functions as a segregation device [57]. Here, water and electrolytes are resorbed, the feces shrink and form pellets, which are covered by a mucus layer (mainly Muc2) [59]. The major form of Muc2 is produced by the proximal colon, which encapsulates the fecal pellets; then a minor form of Muc2 derived from the distal colon leads to a secondary encapsulation [23,24].

The transition from the small to the large intestine is also accompanied by severe histological changes as the villi are only present in the small intestine as well as the Paneth cells [60]. Furthermore, the ascending colon contains transverse folds that mimic human intestinal folds [59,61,62].

The mucus probably plays a key role in microbial colonization of the gastrointestinal mucosa. The rodent stomach and large intestine were described to be covered by a two-layered mucus structure, i.e., a firmly attached, stratified inner layer and a loose outer layer; whereas the small intestine contains only a loose single mucus layer [21]. An inner layer was described in both the proximal and distal colon, but it appeared thicker in the distal part [62]. The inner mucus layer of the murine colon, (probably the distal part [62]), was reported to be devoid of bacteria [63]. However, earlier studies by Swidsinski et al. already reported striking differences between the proximal and distal murine colon [57,64]. They clearly showed that bacteria had direct epithelial contact within the crypts of the proximal colon, but were separated from the distal colonic epithelium; in the latter, no microbiota was detected in the colonic crypts [57,64]. Thus, the mucus in the proximal colon is probably penetrable by bacteria [57]. This view is in line with the model that in the distal colon the bacterial microbiota is confined to the mucus layer of the fecal pellets and thus does not reach the epithelial surface [59]. As a consequence, the empty distal colon is devoid of bacteria [59]. In contrast, in the proximal colon, there is a close contact of the bacterial microbiota with the epithelium in spite of a mucus layer [59].

The intestinal mucus is a secretory product of GCs and MUC2 is the predominant gel-forming mucin. However, intestinal GCs are not a uniform cell population, but they differ at various locations and fulfill different functions. As a hallmark of the colon, rare “sentinel” GCs (senGCs) were identified exclusively at the upper-crypt region, which respond to ligation of specific TLRs, such as TLR2/1, 4, and 5, with inflammasome activation, exocytosis of Muc2, and expulsion of the activated senGCs [65]. Of note, the regional mucosa-associated microbiota determines the expression of TLR2 and TLR4 in the colon and TLR2 is typical of the proximal colon, whereas TLR4 is highest in the distal colon [66]. Thus, there is a reciprocal interaction between microbiota and MUC2 expression. These senGCs are gatekeepers of colonic crypts and they are metabolically linked to other GCs by gap junctions, which allows induction of MUC2 secretion from adjacent GCs [65]. Such a mechanism flushes bacteria away from crypt openings [65]. Furthermore, at least the distal colon contains another sub-population of GCs, i.e., proliferative, non-canonical, and canonical GCs in the crypts as well as “intercrypt” GCs (icGCs) between the crypts [47]. The latter seems to represent fully differentiated canonical GCs [47]. Of special note, canonical GCs in the crypts and icGCs secrete a different mucus; the intercrypt mucus is more permeable to bacteria when compared with the crypt mucus, which seals the crypts by a plume [47]. The two different mucus types can also be distinguished by their lectin binding patterns [47].

Taken together, the special bacterial community in the caecum and ascending colon is probably the result of an intimate relationship of the microbiota with its environment, i.e., the mucus and the luminal content (digesta region), respectively. Hallmarks in the caecum and ascending colon are the dramatic decrease in *Lactobacillus*, and the increase in Bacteroidetes (particularly *Alistipes*), Firmicutes (Lachnospiraceae, unclassified Ruminococcaceae, unclassified Clostridiales), and Deferribacteres (*Mucispirillum*). The latter, typically present in mice but not in humans [56], is found only within the mucus [67] and antagonizes *Salmonella* virulence in mice to protect against colitis [68]. However, *Mucispirillum* does not degrade mucins [69]. The area between the transverse folds of the ascending colon in close apposition to the colonic epithelium was reported to contain mainly Lachnospiraceae and Ruminococcaceae [70]. Thus, a model was proposed concerning the spatial organization of the microbiota across the radial axis of the ascending colon, i.e., large fusiform-shaped bacteria in the mucus (autochthonous/resident microbes), whereas rod- and coccoid-shaped bacteria are associated with the digesta (allochthonous/transient microbes) [61]. This concept was confirmed later for humans [71].

In the transverse and distal colon, particularly Bacteroidetes (unclassified Bacteroidales and unclassified Bacteroidetes) are enriched (Figure 2 and Figure 3). This is an indication for intensive degradation of plant polysaccharides as more than 70% of genes encoding glycoside hydrolases belong to Bacteroidetes members [58].

### 3.2. Comparison of Wild-Type Animals and Tff3^KO^ Mice

In addition to wild-type mice, here also the corresponding *Tff3*^KO^ mice were investigated as they show a high susceptibility in a DSS colitis model due to an intestinal barrier defect; whereas under normal conditions, the *Tff3*^KO^ mice do not show a specific phenotype [29,30]. After pairwise comparison of all the equivalent regions from wild-type and *Tff3*^KO^ animals, PERMANOVA revealed only a weak but significant difference at the genus level in the ascending colon only (*p* = 0.02), whereas all other anatomical locations did not show significant differences between the wild-type and *Tff3*^KO^ mice (data not shown).

This indicates that the presence of Tff3 has no significant influence on the bacterial communities in most regions of the murine intestine. Tff3 is not of advantage or disadvantage for bacterial colonization of the intercrypt mucus. In contrast, Tff3 is obviously important for the protection of the host from bacterial infection during the condition of DSS-induced colitis [29]. Here, Tff3 could protect the sensitive stem cells at the base of the crypts. This hypothesis would be in line with the observation that Tff3 is mainly expressed in the canonical GCs in the crypts, but not in icGCs (drastically reduced Tff3/Fcgbp ratio in the icGCs [47]). Furthermore, DSS treatment caused depletion of senGCs [65]. Thus, it could well be that Tff3—due to its lectin activity—has a function for the viscosity of the protective crypt mucus plume. This assumption would be in line with the report that in humans, TFF3 is positively correlated with the viscoelastic properties of the cervical mucus plug [72].

Thus, in the future, it might be interesting to test whether there is a difference in the bacterial communities particularly within the crypts between wild-type and *Tff3*^KO^ mice during the condition of DSS colitis. In addition, an extension of this study to other microorganisms, such as viruses, fungi, archaea, and protists might detect possible differences between wild-type and *Tff3*^KO^ mice.

### 3.3. Comparison of Gastrointestinal Bacterial Communities in Humans vs. Mice

In the past, the bacterial microbiota was investigated in the human alimentary tract [28] considering biopsies from the gastric antrum and corpus, and the duodenum as belonging to the upper GI tract; whereas biopsies from the terminal ileum and ascending and descending colon were considered as part of the lower GI tract. In this study, we extended the regions, also considering the murine small intestine. We found that the bacterial communities in the murine small intestine are more similar to those in the human upper GI tract than to those in the human lower GI tract. Of note, the major colonizer in the human upper GI tract is *Streptococcus*; whereas in mice it is *Lactobacillus*, both belonging to the phylum Firmicutes. Regarding the lower GI tract, Bacteroidaceae and Lachnospiraceae dominated in both humans and mice (Figure 2 and Figure 3). In humans, the data are limited by an endoscopic biopsy collection (particularly after the duodenum and before the terminal ileum), as well as potential bacterial cross-contamination.

## 4. Materials and Methods

### 4.1. Murine Tissue

Animal experiments were conducted in compliance with the Directive 2010/63/EU of the European parliament and of the council of 22 September 2010 on the protection of animals used for scientific purposes, the German Animal Welfare Act and the regulations on the welfare of animals used for experiments or for other scientific purposes in their currently valid versions. *Tff3*^KO^ and corresponding wild-type mice (mixed 129/Sv and C57BL/6 background) were bred from heterozygous littermates as described previously [73].

The animals were kept in standard cages (IVC Typ II) at the animal facility of the Medical Faculty of the Otto-von-Guericke University Magdeburg under specific-pathogen-free (spf) conditions in a regular 12 h dark-light cycle, maintaining controlled humidity (55 ± 10%) and temperature (22 ± 2 °C) with 15- to 18-fold air exchange rates. Rodent chow diet (ssniff^®^ R/M-H; ssniff Spezialdiäten GmbH, Soest, Germany; gamma-irradiated or autoclaved) and UV-irradiated tap water were provided ad libitum. The animals were not starved before the investigations. Animal care and experimental procedures were performed according to legal regulations (license number: IMMC-TWZ-01). The 12-week-old male mice (9 *Tff3^KO^* animals, weight 19.8–30.6 g; 9 WT animals, weight 22.6–31.4 g) were euthanized by isoflurane inhalation (overdose) and subsequent cervical dislocation.

Samples (stomach: about 0.5 cm in length; all other regions: 1.0–1.5 cm) were taken from 13 regions of the GI tract: esophagus (total); forestomach, gastric corpus, gastric antrum; proximal and distal parts of the duodenum, respectively; middle section of the jejunum, distal ileum, middle part of the caecum; proximal (ascending), medial (transverse) and distal (descending) colon, and rectum. Samples were dissected including the luminal content, immediately frozen in liquid nitrogen and stored at −80 °C.

### 4.2. DNA Extraction and Library Construction

In the first step, samples were suspended in 1 mL of lysis buffer, composed of 100 mM Tris-HCl pH 8.0, 100 mM EDTA, 100 mM NaCl, 1% (*w*/*v*) polyvinylpyrrolidone and 2% (*w*/*v*) sodium dodecyl sulfate, transferred to a 2 mL Lysing Matrix E tube (Qbiogene, Alexis Biochemicals, Carlsbad, CA, USA), subjected to mechanical lysis in a FastPrep^®^ -24 Instrument (MP Biomedicals, Santa Ana, CA, USA) for 40 s and 6.0 m s^−1^, and purified as described [74]. Then, genomic DNA was extracted essentially as previously described [74]. In the final step, amplicon libraries for sequencing were generated as previously described [75], where the V1–V2 region of the 16S rRNA gene was amplified after 20 PCR cycles using 27F and 338R primers [75] and sequenced on a MiSeq (2 × 300 bp, illumina, Hayward, CA, USA).

### 4.3. Bioinformatic and Statistical Analysis

All the fastQ files were analyzed using dada2 package [76] and a unique table containing all samples with the sequence reads and relative abundances was generated. Samples that did not reach 3000 reads (35 samples) were not considered for downstream analysis. Overall, 5,976,799 paired-ends reads were obtained with a mean of 30,339 ± 15,548 reads per sample. Samples with more than 3000 reads (197 samples) were resampled to the minimum sequencing depth of 3172 reads per sample using the phyloseq package [77] returning 4560 phylotypes. The phylotypes were taxonomically annotated using the ribosomal data project based on the naïve Bayesian classification [78] with a pseudo-bootstrap threshold of 80%. Relevant phylotypes were manually annotated until species using NCBI database and the name of species were given if the sequence was identical to the sequence of the correspondent type strain deposited in the database. The vegan package (version 4.0.4) was used to generate the rarefaction curves as well as the phylotypes richness and the Simpson index (1 – λ), where λ = Σp_i_^2^. Relative abundances (in percentage) of phylotypes and genera were used for downstream analyses. Significant differences between locations (esophagus, forestomach, gastric corpus, gastric antrum, proximal and distal parts of the duodenum, respectively, middle section of the jejunum, distal ileum, middle part of the caecum, proximal/ascending, medial/transverse and distal/descending colon, and rectum) were evaluated using permutational multivariate analysis of variance (PERMANOVA) in Past4. Significant differences in the relative abundances between regions were assessed using Mann-Whitney test with Benjamini-Hochberg correction in Prim 7 (GraphPad Software). P values were considered significant if *p* < 0.05 (*p* < 0.05: significant, *; *p* < 0.01: highly significant, **; *p* < 0.001: extremely high significant, ***). Dendrogram was generated using the data matrix, comprising 4560 phylotypes, after the calculation of sample similarity using the Bray-Curtis algorithm.

## 5. Conclusions

In this study, we systematically analyzed the bacterial communities throughout the entire murine gastrointestinal tract. We observed a clear trend in microbiome alterations from the upper, through the middle and through the lower GI tract with its unique microbial composition. The microbial similarity between wild-type and *Tff3*^KO^ mice questions the role of genetics in defining microbial niches under normal conditions. Defined knowledge on microbial trends in the GI tract is crucial for understanding the functional interaction between gut microbes and the host and could prove helpful for further investigations using mouse models. Future systematic studies are needed to address the bacterial microbiota, particularly under the conditions of specific infections, e.g., with *Helicobacter pylori*, or DSS-induced colitis.

## Figures and Tables

**Figure 1 ijms-23-01783-f001:**
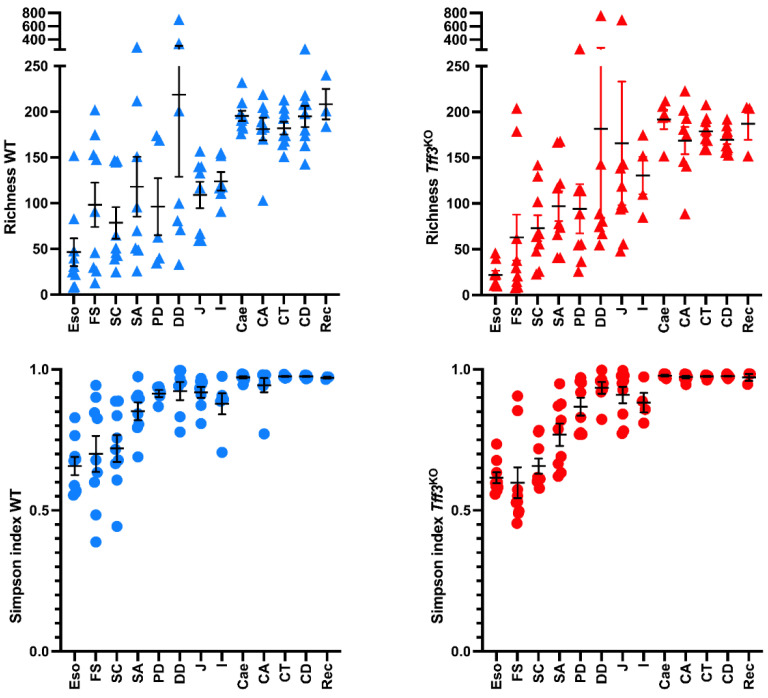
Richness (**top**) and Simpson index (**down**) in wild-type (left panel, in blue) and *Tff3*^KO^ (right panel, in red) mice through the gastrointestinal tract; esophagus (Eso), forestomach (FS), stomach corpus (SC), stomach antrum (SA), proximal duodenum (PD), distal duodenum (DD), jejunum (J), ileum (I), caecum (Cae), ascending colon (CA), transverse colon (CT), descending colon (CD), and rectum (Rec).

**Figure 2 ijms-23-01783-f002:**
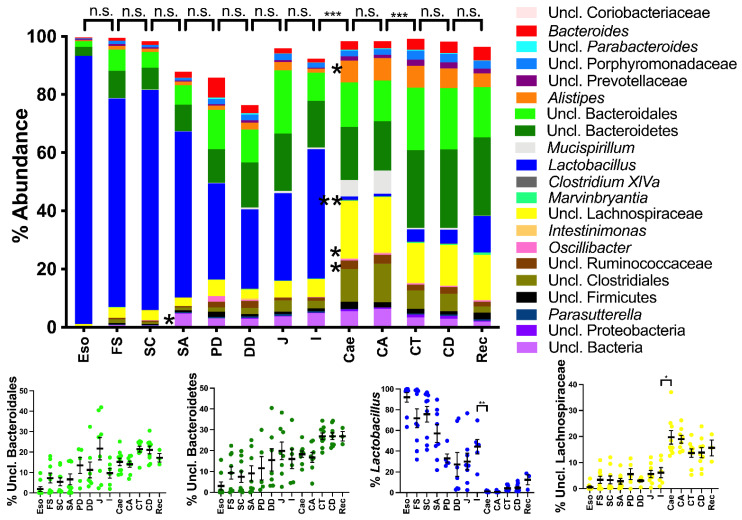
Relative abundance of bacterial microbiota along the alimentary tract of wild-type mice: esophagus (Eso), forestomach (FS), stomach corpus (SC), stomach antrum (SA), proximal duodenum (PD), distal duodenum (DD), jejunum (J), ileum (I), caecum (Cae), ascending colon (CA), transverse colon (CT), descending colon (CD), and rectum (Rec). Top: abundance of taxa detected at least in one region. Pairwise comparison between adjacent regions were calculated with PERMANOVA (n.s., no significant differences; ***, significant differences). Bottom: abundance of taxa belonging to unclassified *Bacteroidales* (light green), unclassified *Bacteroidetes* (dark green), *Lactobacillus* (blue), and unclassified *Lachnospiraceae* (yellow) detected in the different regions. Differences between taxa in adjacent regions were calculated with Kruskal–Wallis test (Benjamin and Hochberg correction); *p* < 0.05: *; *p* < 0.01: **; *p* < 0.001: ***.

**Figure 3 ijms-23-01783-f003:**
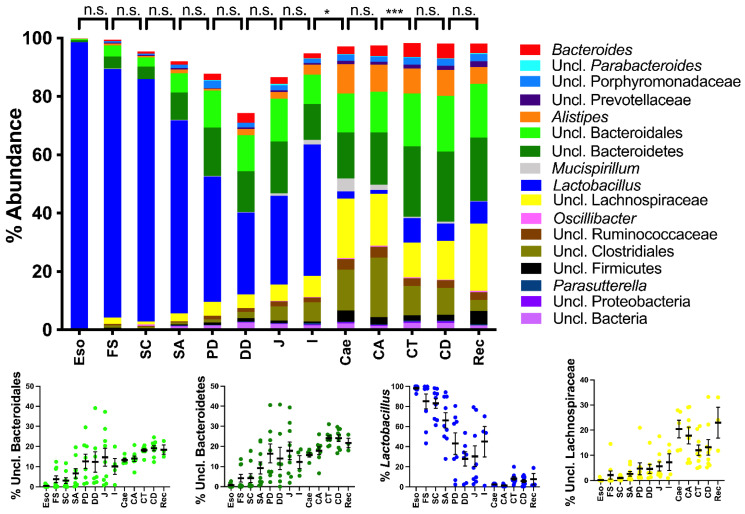
Relative abundance of bacterial microbiota along the alimentary tract of *Tff3*^KO^ mice. For details see the legend of Figure 2.

**Figure 4 ijms-23-01783-f004:**
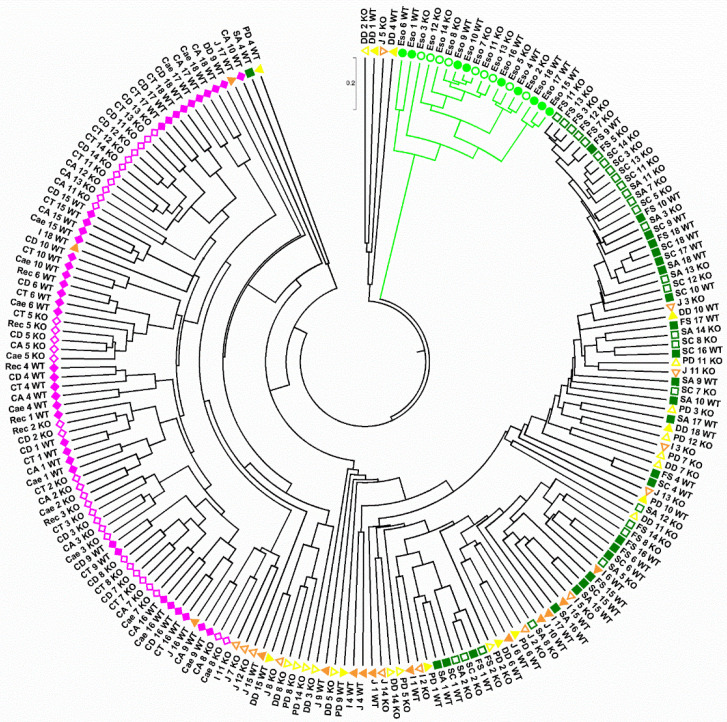
Group-average agglomerative hierarchical clustering of 197 samples, based on the global bacterial profiles at phylotype level and Bray-Curtis similarities. Each sample is termed by its anatomical region (for details see the legend of Figure 2), the number of the animal (from 1 to 18), and the mouse strain (wild-type or *Tff3*^KO^).

**Figure 5 ijms-23-01783-f005:**
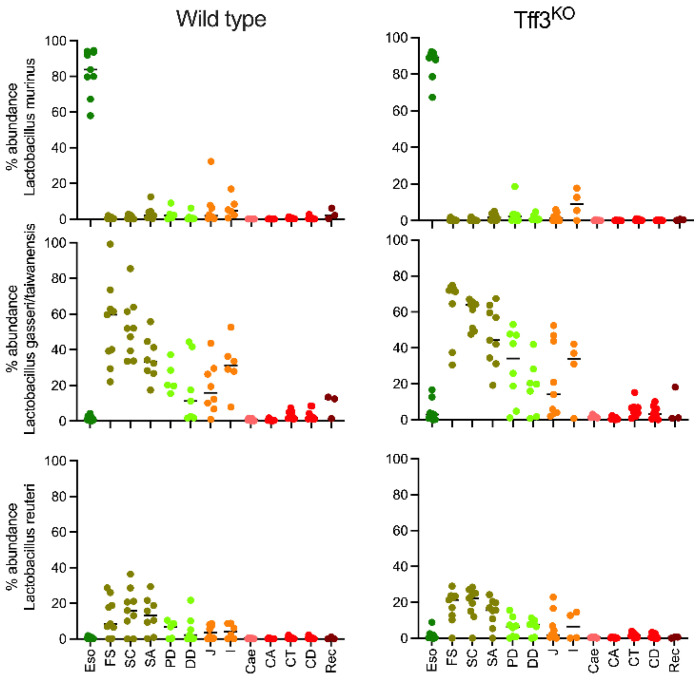
Relative abundance of phylotypes belonging to *L. murinus*, *L. gasseri/taiwanensis* and *L. reuteri* in each mouse and in each region; esophagus (Eso), forestomach (FS), stomach corpus (SC), stomach antrum (SA), proximal duodenum (PD), distal duodenum (DD), jejunum (J), ileum (I), caecum (Cae), ascending colon (CA), transverse colon (CT), descending colon (CD), and rectum (Rec).

## Data Availability

The data presented in this study are available in Appendix A.

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
