# Peer review of "Profiling of the Bacterial Microbiota along the Murine Alimentary Tract"

_ijms, 2022, doi:10.3390/ijms23031783_

Round 1

Reviewer 1 Report

This manuscript reports an interesting study on the gut microbiota communities differentially distributed in the GI tracts of mice. The authors provided a comprehensive analysis of the bacterial communities in 13 anatomical regions from the esophagus to the rectum. They demonstrated a clear trend in microbiome shift from the upper through the lower GI tract with unique microbial composition. However, this study is largely descriptive without functional validation and characterization of differential bacterial species identified in this study.  There are several concerns listed below.

  1. The rationale for studying gut microbiota dysbiosis in Tff3-deficient (Tff3 KO) mice is unclear since wild-type and Tff3 KO animals did not show significant differences in their bacterial communities. Further justification to study Tff3 KO mice is warranted as Tff3 KO indeed develops intestinal barrier dysfunction, leading to gut microbiota dysbiosis.
  2. This study lacks novelty as bacterial diversity across the GI tract is already known. Further, there is no attempt to discuss the functional relevance of the gut microbiota in different GI sections. Functional characterization of the bacterial population in different GI sections is warranted. How do the host-gut microbiota and/or their derived metabolites interact each other and modulate the gut physiology? It is important to clearly address direct and/or indirect interactions between host and gut microbiota.
  3. Translational validation of the bacterial communities is missing. The authors should validate the findings in human biopsies and/or stool samples.
  4. How is the gut microbial community correlated with the gut immune phenotypes in Tff3 KO mice? Did the authors attempt to elucidate gut immune dysfunction in this mouse model? 
  5. The authors need to address the limitations of the study.
  6. The discussion section needs extensive revision—particularly the literature on the commensal/pathogenic bacteria in different GI sections modulating the physiology and gut immune dysfunction.

Author Response

Reviewer 1

Ad 1. The rationale for studying potential gut microbiota dysbiosis in Tff3KO mice is well explained in the introduction (lines 77-91). However, we further expanded this point in the introduction (lines 93-93) and the discussion (lines 212-214). Generally, with this study we wanted to set the base line for further studies.

Ad 2. We disagree with this comment. For example, it is new that the genus Lactobacillus forms the major bacterial community from the esophagus to the ileum. In our opinion it is the first study, where a systematic and detailed mapping of the bacterial community along the entire murine GI tract is shown. It is not the aim of this study to analyze the host-gut interactions and their metabolites.

Ad 3. The only study mapping systematically the bacterial community in humans was reported by Vasapolli et al. (cited as ref. 28). We compare the situation in mice versus human even in an own section (Section 3.3). For example, Lactobacilllus is present in mice; whereas Streptococcus is present in human and we also mention that Bacteroidetes are present in both.

Ad 4. The elucidation of a gut immune dysfunction was not the aim of this study. We wanted to state the base line for further studies by comparing wild type andTff3KO mice (feeding the same diet).

Ad 5. Thank you for this comment. One limitation might be that the databases for taxonomical annotations of bacterial communities are more oriented to human experiments rather than for a mouse model (we limited the study to the region V1-V2 of the 16S rRNA gene). Thus, we were able to annotate a reasonable amount of phylotypes only to the family or even less resolution taxonomy level. For a functional characterization, a metagenomic/metatranscriptomic approach would have been needed, which was not the aim of the current study.

Ad 6. We focused on bacterial communities previously described in mice in order to compare our results. Our current study was oriented to set a base line without any stress on both mouse strains (wild type and Tff3KO and no pathogenic bacteria).

Reviewer 2 Report

In my opinion the manuscript (ID ijms-1536909) entitled “Profiling of the Bacterial Microbiota along the Murine Alimentary Tract” by Ramiro Vilchez-Vargas, Franz Salm, Eva B. Znalesniak, Katharina Haupenthal, Denny Schanze, Martin Zenker, Alexander Link and Werner Hoffmann is very interesting and should be published in International Journal of Molecular Sciences after only minor revision.

Authors did a huge work and determined the spatial distribution of the bacterial community along the murine alimentary tract (13 different regions from the esophagus to the rectum). They analyzed wild type animals and Tff3-deficient (Tff3KO) mice, which are useful in studying gastrointestinal tract injury (including inflammatory bowel diseases). What is important, the Authors examined the number of bacterial species (richness) as well as the diversity of bacterial communities present in the murine GI.

I have only few remarks:

  1. The most important concerns the Simpson index used in the study. Since the mean proportional abundance of the types increases with decreasing number of types and increasing abundance of the most abundant type, the Simpson index λ obtains small values in datasets of high diversity and large values in datasets of low diversity. This is counterintuitive behavior for a diversity index, so often, such transformations of λ that increase with increasing diversity have been used instead. The most popular of such indices have been the inverse Simpson index (1/λ) and the Gini–Simpson index (1 − λ). As noted in the manuscript, the higher diversity, the higher was the Simpson index. Could you provide therefore (in Material and Methods chapter) the formula used to calculate the Simpson index in the paper?
  2. Figure 2. What mean the asterisks (** or *) inside the Figure 2 (e.g., between bars I and Cae)? I suppose that Authors wanted to present the significant differences in particular taxa, but this is not described clearly. Maybe the asterisks should be inside the bar Cae, in the location of appropriate color representing the taxa with significant difference? Did the differences in Mucispirillum abundance in I and Cae were significant? Were any other differences statistically significant? Eg. yellow bar (Lachnospiraceae) between Eso and FS? The same questions is for unclassified Ruminococcaceae and Oscillibacter between adjacent regions SA and PD?
  3. Line 164 and Fig. 2 - Are you sure that there is no statistical differences between DD and J, wheras the difference between I and CAE is significant at p<0.001(***)?
  4. Please, write the names of bacterial genus and species in italics (eg. chapter 2.3 has many non-italics)
  5. Line 113. Correct the title of chapter - the KO in Tff3KO mice should be written as superscript

Author Response

Reviewer 2

Thank you for your nice comments.

Ad 1. Thank you for your suggestion. We definitely applied Gini-Simpson (1 – λ). Simpson’s index is based on λ = Σpi2. We included this formula also in Section 4.3.

Ad 2. We regret the misunderstanding. We calculated the differences between taxa in adjacent regions using the Kruskal-Wallis test with BH corrections and we only presented those with statistically significant differences. Mucispirillum and all the others were not significant due to the high variance within mice. Certainly, some mice showed high abundance of Mucispirillum and all the other taxa, but those taxa were not present in sufficient number of mice in order to be significant (a detailed percentage of abundancies is shown in Supplementary file 1).

Ad 3. Indeed, between DD and J there is not a statistically significant difference using PERMANOVA, most probably because of the variance between the mice. We only show those pairwise comparisons, which had statistically significant differences.

Ad 4. Thank you for this comment. We checked that.

Ad. 5. Thank you for this comment. This was corrected.

Round 2

Reviewer 1 Report

Although the authors made some helpful revisions, the revisions are incremental without addressing the fundamental concerns below.

  1. Although the authors provided the rationale for using the Tff3 KO mouse model in this revision, the underlying mechanism of the gut microbial community linked with the gut immune parameters is not demonstrated yet. The authors said, this was not the study's aim, however, addressing the gut immune dysfunction associated with gut bacteria is critical for this study.
  2. Unfortunately, the authors did not provide any data for the mucosal barrier integrity function. At least discussion is warranted to link the gut bacteria to intestinal barrier dysfunction and to address how the study results add new information to the existing data in literature.
  3. Finally, how the unique gut microbe-derived metabolites interaction with the host to modulate physiological gut functions in the specific regions of the GI tract should be discussed and addressed clearly.

Author Response

Ad 1. Underlying mechanism: As already stated in the introduction and the discussion as well, there are multiple indications that TFF3, together with FCGBP, plays a role in the innate immune defence of mucous epithelia, probably by strengthening the outer colonic mucus barrier (discussed in detail in ref. [45]).

Furthermore, TFF3 expression is linked to inflammation (review: [30]) and also a complex immunomodulation of goblet cells is well known, which has been investigated in murine model systems. For example, Tff3 expression is reduced by the immune system after infection with Citrobacter rodentium, which is not observed in Rag1KO mice [Bergstrom et al. 2008]. This could be explained by a link of T lymphocytes and homeostasis of intestinal epithelial cells by IL-7 [Shalapour et al. 2012]. Furthermore, Tff3 expression is induced by TLR2 activation by commensal bacteria probably by an indirect mechanism [Podolsky et al. 2009]. These data are NOT discussed in this manuscript as they are not directly related to the topic investigated here. However, we introduced now a short statement pointing to the complex immunomodulation of goblet cells (discussion: lines 215-221 plus 3 more references).

In this manuscript, the role of Tff3 as a constituent of the intestinal mucus is of relevance; and Tff3KO mice have a partially impaired mucus barrier, particularly in the DSS colitis model [31, 45].

Ad 2. Thus far, the details of the mucosal barrier integrity function of TFF3 are not known, e.g. how TFF3 and/or FCGBP interact with bacteria etc. However, there are multiple indications that TFF3 is part of the mucosal innate immune defence [45]. This has been clearly mentioned in the introduction as well as the discussion. Generally, this study does not add new data to this point.

Ad 3. As we already stated in the cover letter for the first revision, this study is NOT designed to analyse the host-gut interactions and their metabolites. Although alterations of the microbiome may be considered as surrogate for the metabolite’s changes, it does not necessarily reflect the absolute changes. For the future, a metabolites atlas of the GI mucosa would be very valuable but also very challenging and is not within the scope of this manuscript.

Round 3

Reviewer 1 Report

I have no further concerns or comments.